# Fast Logic Function Extraction of LUT from Bitstream in Xilinx FPGA

**Soyeon Choi and Hoyoung Yoo *** 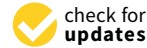

Department of Electronics Engineering, Chungnam National University, Daejeon 34134, Korea;
soyeonchoi@cnu.ac.kr
***** Correspondence: hyyoo@cnu.ac.kr

**Abstract:** This paper presents a fast method to extract logic functions of look-up tables (LUTs) from a bitstream in Xilinx FPGAs. In general, FPGAs utilize LUTs as a primary resource to realize a logic function, and a typical $N$-input LUT comprises $2N$ 1-bit SRAM and $N - 1$ multiplexers. Whereas the previous research demands $2^N$ exhaustive processing to find a mapping rule between an LUT and a bitstream, the proposed method decreases the processing to $2N$ by eliminating unnecessary processing. Experimental results show that the proposed method can reduce reversing time by more than 57% and 85% for Xilinx Spartan-3 and Virtex-5 compared to the previous exhaustive algorithm. It is noticeable that the reduction time becomes more significant as a commercial Xilinx FPGA tends to include a more tremendous number of LUTs.

**Keywords:** look-up-table (LUT); reverse engineering; bitstream; FPGA; Xilinx

## 1. Introduction

Field programmable gate arrays (FPGAs) are a type of semiconductor device that can be reconfigured by register transfer level (RTL) designers to realize target functionality. A typical FPGA conceptually consists of a tile of three major blocks [1,2], configurable logic blocks (CLBs), input/output blocks (IOBs), and switch matrices (SMs), as shown in Figure 1. CLB is a primary resource to realize target logic function. Each CLB is decomposed into several SLICEs, each of which contains look-up-tables (LUTs), flip-flops (FFs), and multiplexes (MUXs). In addition, IOBs are responsible for controlling external connectivity and SMs provide configurable internal connectivity between the CLBs and IOBs within the FPGA. It is important that the configurability of FPGA originates from values that are stored in programmable points [1,2] such as programmable logic points (PLP), programmable interconnect points (PIP), and programmable content points (PCP), denoted as bold red boxes in Figure 2. According to the values stored in the programmable points, a FPGA is allowed to provide various logical functionalities as required by the RTL designer intends. Due to the programmable nature, FPGAs have been widely adopted for many fields of the embedded systems, including consumer electronics [3], communication systems [4], automotive vehicles [5], and defense industry applications [6].

Among different types of FPGAs, including anti-fuse-based [7,8] and Flash-based [9,10], SRAM-based [11] FPGAs have dominated the market owing to their high density, low cost, and fast configuration time [12]. One weakness of SRAM-based FPGAs is that it essentially requires an external nonvolatile memory to store a netlist because SRAM is a type of volatile memory. It is inevitable that the bitstream stored in the external nonvolatile memory should be transferred to the SRAM-based FPGA whenever the FPGA system is powered on. To protect the bitstream from malicious attackers, most FPGA manufacturers have supported bitstream encryption [13], but they have also attempted to

decrypt the encrypted bitstream by estimating encryption keys [6]. In this manuscript, we assumed that the bitstream is not encrypted to clarify the reverse engineering process.

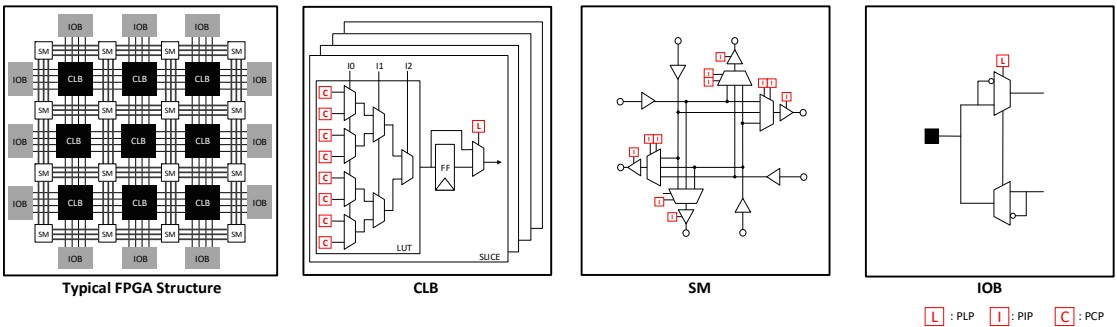

**Figure 1.** Typical FPGA structure.

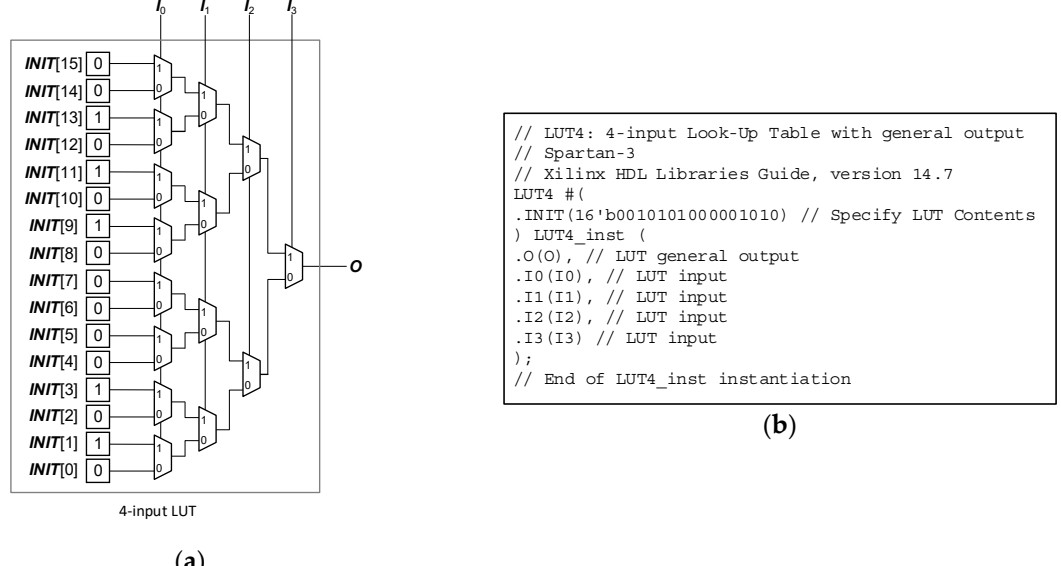

**Figure 2.** (**a**) Structure and (**b**) primitive library for 4-input LUT.

Previously, many studies about reverse engineering [14–23] have tried to recreate the original design after extracting the bitstream from the external memory while transferring it from a nonvolatile memory. Xilinx supports two integrated development environment—Xilinx ISE Design Suite and Vivado—to synthesize, simulate, and program FPGA chips. Xilinx ISE Design Suite supports low-cost FPGA chips, including Spratan-6 and Virtex-6, as well as their previous families, and Xilinx Vivado supports high-performance FPGA chips, including state-of-the-art Virtex-7, Kintex-7, and Artiex-7. Most previous studies have focused on recovering bitstreams generated from Xilinx ISE Design Suite according to [14–21], and recent studies in [22,23] have started to investigate recovering bitstream generated from Xilinx Vivado. Since Xilinx ISE Design Suite has continued to be widely used to support various types of low cost FPGA chips, reverse engineering using Xilinx ISE Design Suite is as important as using Xilinx Vivado. Since reverse engineering tries to extract the original design from bitstream, it results in security issues when reverse engineering is abused. On the other hand, reverse engineering can provide an efficient security solution for an ethical purpose when malicious modifications like hardware Trojan [24–28] are infiltrated into the original circuits. More precisely, reverse engineering can detect malicious modification by comparing the regenerated design form bitstream and the original netlist. Many previous researches, including [19,20], have discussed this security issues and concerns.

Since the essential information of programmable points are included in the extracted bitstream, reverse engineering for FPGAs is considered as the process to reconstruct a mapping rule between bitstream and programmable points for the target FPGA. Many previous researches [14–21] have focused on examining a mapping rule for PLPs and PIPs and succeeded in recovering the mapping rule at a high level of accuracy. However, there are seldom researches to discover PCPs represented as LUTs [6,29], and an efficient method to restore PCPs in terms of both accuracy and speed is needed. For the first time, the authors of [6] presented a method to resynthesize a logic function by exhaustively searching all possible initial values targeting for Xilinx FPGAs. More precisely, each $N$-input LUT is synthesized $2^N$ times with different initial values. For instance, to restore one 4-input LUT, the LUT is needed to synthesize 18 times with different initial values. Although the number of input $N$ is normally small for FPGA, the total recovering time in a current FPGA becomes no longer negligible due to the huge number of LUTs. As an example, the total recovering time in [6] becomes 102 weeks for Xilinx Virtex-5 having 19,200 LUTs since it demands 53 min for one LUT if the processing time for one synthesis is 43 s. As a result, exhaustive searching [6] should be improved, especially for an advanced FPGA with a tremendous number of LUTs.

To mitigate this problem, this paper presents a fast method for logic function extraction of $N$-input LUT by decreasing the number of synthesis from $2^N$ to $2N$. Unnecessary synthesis is completely eliminated without affecting restoring accuracy, which changes exponential increase to linear increase to consequently reverse complexity. The rest of this paper is organized as follows: Section 2 describes the backgrounds, including an LUT structure and the details of the previous exhaustive method [6]. Section 3 explains the proposed fast extraction method focusing on how the number of synthesis can be reduced. Section 4 discusses experimental results using Xilinx Spartan-3 and Virtex-5, followed by concluding remarks in Section 5.

## 2. Background

### 2.1. LUT Structure

In general, FPGA employs LUTs as a primary resource to realize a target logic function. A typical $N$-input LUT consists of $2^N$ 1-bit SRAM cells and $N - 1$ 2-to-1 MUXs. Figure 2a depicts 4-input LUT as an example with 16 1-bit SRAM cells and 15 2-to-1 MUXs. Based on the structure of 4-input LUT, $INIT[I]$ associated with the MUX input $I = 4'bI_3I_2I_1I_0$ is selectively determined as the output $O$. For instance, when the input $I$ sets to 4'b0110 in Figure 2a, the output becomes 1 corresponding to $INIT$ [6]. The vector form of the input is also represented as the Boolean product form by setting each $i$-th Boolean variable as either $I_i$ or $\overline{I_i}$ depending on whether the $i$-th bit pattern is 1 or 0, respectively. The input $I = $ 4'b0110 can be represented as $\overline{I_3}I_2I_1\overline{I_0}$ additionally.

Using Boolean product form, we can determine all bits in $INIT$ for a specific logic function. Let us assume that the logic function is $O = I_3\overline{I_1}I_0 + \overline{I_2}I_0$. First, the Boolean function should be expanded as the standard form that includes each input variable in all the product terms. The standard form is easily obtained by applying well-known distribution rule, $I_iI_j + I_iI_k = I_i(I_j + I_k)$ and complement rule $I_i + \overline{I_i} = 1$. The standard form of the target logic function $O$ is computed as

$$
\begin{aligned}
O &= I_3\overline{I_1}I_0 + \overline{I_2}I_0 \\
&= I_3(I_2 + \overline{I_2})\overline{I_1}I_0 + (I_3 + \overline{I_3})\overline{I_2}(I_1 + \overline{I_1})I_0 \\
&= I_3I_2\overline{I_1}I_0 + I_3\overline{I_2}\overline{I_1}I_0 + I_3\overline{I_2}I_1I_0 + I_3\overline{I_2}\overline{I_1}I_0 + \overline{I_3}\overline{I_2}I_1I_0 + \overline{I_3}\overline{I_2}\overline{I_1}I_0 \\
&= I_3I_2\overline{I_1}I_0 + I_3\overline{I_2}I_1I_0 + I_3\overline{I_2}\overline{I_1}I_0 + \overline{I_3}\overline{I_2}I_1I_0 + \overline{I_3}\overline{I_2}\overline{I_1}I_0
\end{aligned}
\tag{1}
$$

Based on the relation between the vector and Boolean forms, the content of LUT can be obtained as 16'b0010_1010_0000_1010. As a result, any Boolean logic function with 4 input variables can be configured in a 4-input LUT by storing suitable values in the SRAM.

Furthermore, Xilinx provides primitive libraries [30,31] of LUTs written in hardware description language (HDL) to help RTL designs to instantiate LUT. Figure 2b depicts the primitive library of

4-input LUT with the name of LUT4, the inputs of $I_3$, $I_2$, $I_1$, $I_0$, the output of $O$, and the initial value of 16-bit $INIT$. The LUT contents computed from the target logic function is initialized as a 16-bit $INIT$ parameter. When $INIT$ is initialized as 16'b0010_1010_0000_1010, the 4-input LUT operates the logic function $O = I_3\overline{I_1}I_0 + \overline{I_2}I_0$.

Reverse engineering seems straightforward if the LUT content identical to the $INIT$ value is explicitly shown in the extracted bitstream since it can be easily converted using the relation between logic function and bit patterns, as described previously. However, the LUT content is not explicitly shown in the extracted bitstream unfortunately. In fact, many FPGA manufacturers, including Xilinx [32] and Intel [33], obfuscate the LUT content to protect its original design from IP theft and prevent malicious manipulation. The 4-input LUT is actually represented as Figure 3 in Xilinx Spartan-3 rather than Figure 2a. When an RTL designer instantiates a LUT with $INIT$ bits targeting a specific logic function, the RTL designer is interested in 16-bits of $INIT$ associated with the input variables of $I_0$, $I_1$, $I_2$, and $I_3$. However, $INIT$ are not directly used to fabricate the commercial FPGA chips, which means that $INIT[i]$ is not explicitly shown in the actual bitstream. Instead of 16-bits of $INIT$ and input variables of $I_0$, $I_1$, $I_2$, and $I_3$, alternative signals denoted as 16-bits of $BIT$ and input variables of $A_1$, $A_2$, $A_3$, and $A_4$ are internally used to provide secure operation. As shown in Figure 3, 16-bit $INIT$ is translated as 16-bit $BIT$ rather than using 16-bit $INIT$ directly in bitstream. Moreover, the input variables of $I_0$, $I_1$, $I_2$, and $I_3$ are translated as $A_1$, $A_2$, $A_3$, and $A_4$, used in synthesis and implementation process in Xilinx design suits. Note that we follow the index of $A_j$ for $1 \leq j \leq N$ as used in Xilinx design suits without loss of generality. As a result, the reverse engineering seeks to disclose the mapping rule between $INIT$ and $BIT$ vectors. Since a bitstream includes all $BIT$ vectors corresponding to all LUTs in a target FPGA, reverse engineering can be successful by translating each bit $BIT$ vector to $INIT$ vector and converting a logic function from the translated $INIT$ vector when a mapping rule for each LUT is completely recovered.

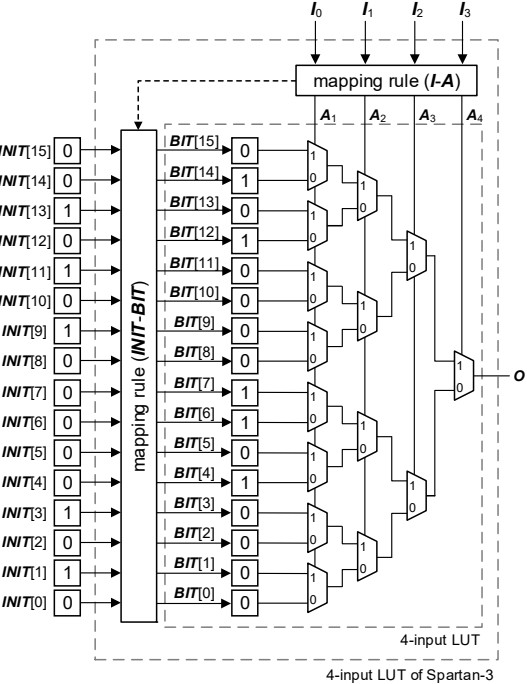

**Figure 3.** Actual 4-iput LUT structure in Xilinx Spartan-3.

## 2.2. The Previous LUT Extract Method

Since a bitstream can be easily extracted while transferring from an external memory to FPGA by using a measurement equipment like a logic analyzer, we assume that the bitstream extraction

is successfully accomplished, and thus focus on the discovery of the mapping rule. Swierczynski in [6] proposed the method to discover the mapping rule between ***INIT*** and ***BIT*** for the first time. The previous exhaustive processing [6] mainly demands three steps as follows.

### 2.2.1. Bit Position Search

The bit position for a target $N$-input LUT is searched since there is no information about the bitstream. To find the position of bits associated with the target LUT, one target $N$-input LUT is instantiated two times with ***INIT*** vectors of $2^N$-bit all zeros and ones using primitive libraries in Figure 2b. Note that the other hardware resource, except for the ***INIT*** vectors, maintain the same. From the comparison after synthesizing two designs using Xilinx ISE Design Suite, it is found that the two generated bitstreams are exactly different by $2^N$ bits equal to the length of ***INIT***. As an example, 16 different positions are obtained for 4-input LUT by implementing the ***INIT*** with 16-bit zeros and ones.

### 2.2.2. Mapping Table Construction

The same $N$-input LUT is instantiated $2^N$ times with different ***INIT*** vectors maintaining the other hardware resources the same. For the $i$-th instantiation, the LUT is initialized with the ***INIT*** vector with single one on the $i$-th position and all zeros on the other positions. After synthesizing and implementing the target LUT, Xilinx ISE Design Suite generates a bitstream and the $N$-bit pattern denoted as ***BIT*** is extracted from the previously selected bit positions. The exhaustive $2^N$ instantiation results in $2^N$ ***INIT*** and $2^N$ ***BIT*** vectors for the target LUT. Given $2^N$ pairs of ***INIT*** and ***BIT*** vectors, the mapping table is finally constructed by extracting the position of single one from the pairs of vectors. Figure 4 exemplifies the construction of the mapping table for 4-input LUT. For instance, when ***INIT*** sets to 16'b0000_0000_1000_0000, ***BIT*** becomes 16'b0010_0000_0000_0000 through Xilinx synthesis and an implementation process, and the relation between ***INIT*** [7] and ***BIT*** [13] is obtained. From the exhaustive 16 processing, all elements in the mapping table can be completely filled.

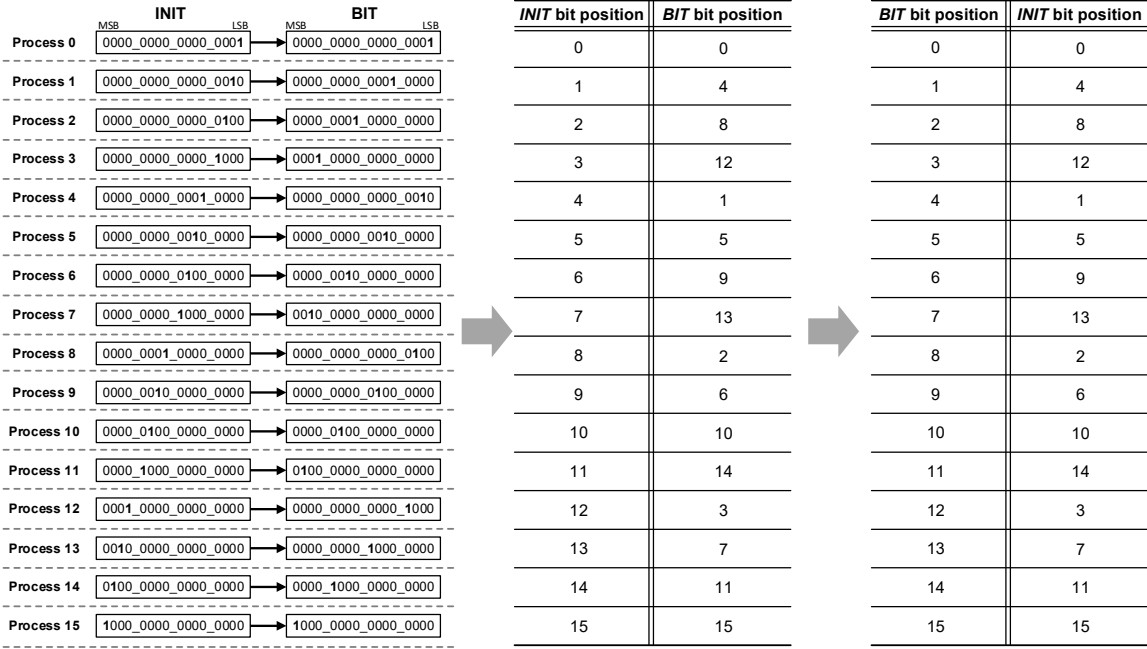

**Figure 4.** Mapping table construction for the previous method [6].

Lastly, it is noticeable that ***BIT*** for all Xilinx FPGAs does not always has single one vector although the ***INIT*** is initialized with single one vector. As an example, ***BIT*** for Xilinx Spartan-3 is single zero

vector whereas **BIT** for Xilinx Virtex-5 is single one vector. For instance, **INIT** 16′b0000_0000_1000_0000 in Xilinx Spartan-3 results in **BIT** 16′b1101_1111_1111_1111 instead of 16′b0010_0000_0000_0000. As **BIT** is either single one or zero vector, the mapping table can be seamlessly constructed if **BIT** inversion checks once.

### 2.2.3. Logic Restoration

After constructing the mapping table, the final step to restore a logic function is straightforward. First, the $2^N$ bit vector denoted as **BIT** is extracted from the bitstream based on the selected bit positions of Section 2.2.1. The constructed mapping table of Section 2.2.2 allows to convert a $2^N$ **BIT** vector to a $2^N$ **INIT** vector. Since **INIT** vector is the content of the LUT, **INIT** vector is translated as the LUT inputs of $I_0$, $I_1$, $I_2$, and $I_3$ according to Boolean representation. As an example, assume that the target **BIT** is 16′b0101_0000_1101_0000 extracted from the target bitstream at the selected bit position of bit position search in Section 2.2.1. Once **BIT** vector is acquired, **INIT** is simply obtained as 16′b0010_1010_0000_1010 by rearranging **BIT** based on the mapping table shown in Figure 4. Using the Boolean representation, **INIT** [1], **INIT** [3], **INIT** [9], **INIT** [11] and **INIT** [13] in **INIT** vector are converted as $\overline{I_3 I_2 I_1} I_0$, $\overline{I_3 I_2} I_1 I_0$, $I_3 \overline{I_2 I_1} I_0$, $I_3 \overline{I_2} I_1 I_0$ and $I_3 I_2 \overline{I_1} I_0$. Finally, the logic function $O = I_3 I_2 \overline{I_1} I_0 + I_3 \overline{I_2} I_1 I_0 + I_3 \overline{I_2 I_1} I_0 + \overline{I_3 I_2} I_1 I_0 + \overline{I_3 I_2 I_1} I_0$ is restored from **BIT** 16′b 0101_0000_1101_0000.

The previous method with $2^N$ exhaustive processing has succeeded in restoring a logic function of a LUT from the bitstream. However, it is impractical in that a FPGA generally contains more than hundreds and thousands of LUTs, and thus the number of processing increases exponentially when the previous exhaustive method [6] is applied. Therefore, it is necessary to improve the previous method to provide a practical reverse engineering solution for LUTs.

## 3. Proposed Method

In this paper, we present a new fast method to reduce recovery time by saving the number of processing. Whereas the previous exhaustive method [6] to perform reverse engineering on an $N$-input LUT requires $2^N$ processing exponential to $N$ to perform reverse engineering on an $N$-input LUT, the proposed method requires $2N$ processing linear to $N$. The main idea of the proposed method is to employ an individual basis of $I_i$ ($0 \leq i < N$), not the entire vector form of $I = I_{N-1} I_{N-2} \dots I_1 I_0$. For the case of 4-input LUT, as an example, the previous exhaustive method [6] generates 16 **BIT** vectors from all possible **INIT** vectors from $I = 0$, i.e., $I_3 I_2 I_1 I_0 = 0000$, to $I = 15$, i.e., $I_3 I_2 I_1 I_0 = 1111$. However, the proposed method generates 4 **BIT** vectors according to individual input $I_0$, $I_1$, $I_2$, and $I_3$, and the 4 **BIT** vectors are used as a basis to construct a mapping table.

The generation of $2^N$-bit **BIT** vectors for individual input $I_i$ ($0 \leq i < N$) seems straightforward. The individual input $I_i$ ($0 \leq i < N$) seems possible to be synthesized and implemented using $N$-input LUT primitive library initialized with an appropriate $2^N$-bit **INIT** vector, as the vector form of $I = I_{N-1} I_{N-2} \dots I_1 I_0$ is instantiated in the previous method. The **INIT** vector can be computed using Boolean distribution and complement rules. For instance, **INIT** value of $I_0$ is 16′b1010_1010_1010_1010 since

$$
\begin{aligned}
I_0 &= I_0 (I_1 + \overline{I_1})(I_2 + \overline{I_2})(I_3 + \overline{I_3}) \\
&= I_3 I_2 I_1 I_0 + I_3 I_2 \overline{I_1} I_0 + I_3 \overline{I_2} I_1 I_0 + I_3 \overline{I_2 I_1} I_0 + \overline{I_3} I_2 I_1 I_0 + \overline{I_3} I_2 \overline{I_1} I_0 + \overline{I_3 I_2} I_1 I_0 + \overline{I_3 I_2 I_1} I_0
\end{aligned}
\tag{2}
$$

This attempt to use the LUT primitive library with a single input seems logical, but it is actually impossible to generate a **BIT** vector due to practical reasons. While synthesizing and optimizing the LUT, the Xilinx ISE Design Suite generally eliminates the LUT instantiated with a single input. Instead of the LUT instantiation, the single input is directly bypassed to the output to save hardware resource. Note that as far as we know, there is no feasible way to synthesize and implement an LUT primitive library with a single input using Xilinx ISE Design Suite.

### 3.1. *I-A* Mapping Construction

To solve this problem, the proposed method employs Xilinx Design Language (XDL) [34,35], which describes the utilized hardware components for the current target design among the entire hardware resources in FPGA. In Xilinx ISE Design Suite, XDL [34,35] is generated automatically to provide the RTL designers for verifying the target design's synthesis and optimization. Figure 5 shows the example of XDL [34,35] captured from Xilinx Spartan-3. The red box in XDL [34,35] indicates that the current design instantiates LUT4 whose internal Boolean function is $(A_3((\sim A_4 * A_2) + \sim A_1))$ when synthesizing a 4-input LUT with target function $I_3\overline{I_1}I_0 + \overline{I_2}I_0$. As previously descried in Section 2, the actual configuration of LUT in Figure 3 is not the same as the ideal LUT in Figure 2a.

```
inst "O_OBUF" "SLICEL",placed R1C1 SLICE_X0Y94  ,
  cfg " BXINV::#OFF BYINV::#OFF CEINV::#OFF CLKINV::#OFF COUTUSED::#OFF
       CY0F::#OFF CY0G::#OFF CYINIT::#OFF CYSELF::#OFF CYSELG::#OFF DXMUX::#OFF
       DYMUX::#OFF F::#OFF F5USED::#OFF FFX::#OFF FFX_INIT_ATTR::#OFF FFX_SR_ATTR::#OFF
       FFY::#OFF FFY_INIT_ATTR::#OFF FFY_SR_ATTR::#OFF FXMUX::#OFF FXUSED::#OFF
       G:LUT4_inst:#LUT:D=(A3*((~A4*A2)+~A1))) GYMUX::G REVUSED::#OFF SRINV::#OFF
       SYNC_ATTR::#OFF XBUSED::#OFF XUSED::#OFF YBUSED::#OFF YUSED::0 "
  ;
```

**Figure 5.** Example of XDL in Spartan-3.

Since an LUT cannot be implemented with a single input $I_i$ ($0 \leq i < N$) due to practical reasons, the proposed method instantiates a specific form of $I = I_{N-1}I_{N-2} \ldots I_1I_0$ that can distinguish the relation between input variable $I_i$ ($0 \leq i < N$) and internal variable $A_j$ ($1 \leq j \leq N$). More precisely, $I$ sets to the combination of single $I_i$ and other $\overline{I_i}$. For instance, 4-input LUT is instantiated with $\overline{I_3I_2I_1}I_0$, $\overline{I_3I_2}I_1\overline{I_0}$, $\overline{I_3}I_2\overline{I_1I_0}$ and $I_3\overline{I_2I_1I_0}$. Using this specific form of $I$ and corresponding XDL [34,35], a feasible solution is provided to map input variable $I_i$ ($0 \leq i < N$) and internal variable $A_j$ ($1 \leq j \leq N$). As an example, Figure 6 shows that input $I$ of $\overline{I_3I_2I_1}I_0$, $\overline{I_3I_2}I_1\overline{I_0}$, $\overline{I_3}I_2\overline{I_1I_0}$ and $I_3\overline{I_2I_1I_0}$ are initialized in the target LUT with a different 16-bit *INIT* vector, resulting in different XDLs. Based on the input $I$ and XDL [34,35] described in Figure 6, pairs of ($I_0$, $A_3$), ($I_1$, $A_4$), ($I_2$, $A_1$), and ($I_3$, $A_2$) are recovered. Thus, $N$-input LUT demands $N$ processing to completely build the *I–A* mapping table.

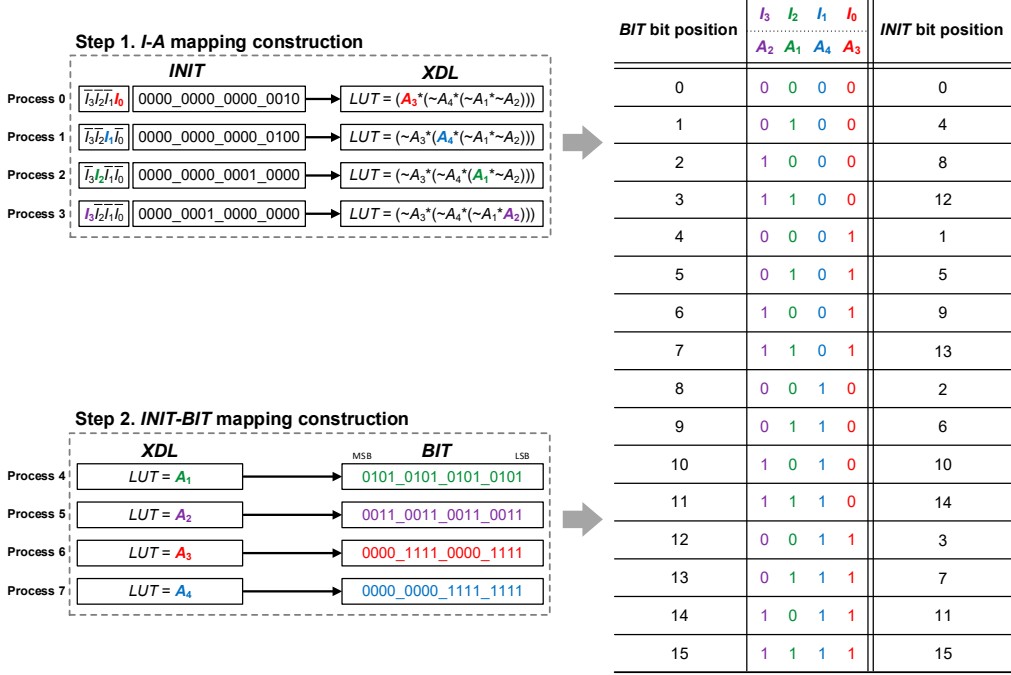

**Figure 6.** Mapping table construction for the proposed method.

*3.2. **INIT-BIT** Mapping Construction*

　　To recover programmable points such as PLP and PIP, many previous researches [14–21] employ XDL [34,35] to modify the original design as intended. In the proposed method, XDL [34,35] is modified to instantiate an LUT with a single internal variable $A_j$ ($1 \leq j \leq N$). Since synthesis with a single input $I_i$ ($0 \leq i < N$) is practically infeasible, we synthesize LUTs with a single internal variable $A_j$ ($1 \leq j \leq N$) alternatively. Using the mapping relation between $I_i$–$A_j$ obtained at the previous step, the generated **BIT** associated with an internal variable $A_j$ ($1 \leq j \leq N$) is rearranged to restore a mapping table. Figure 6 depicts the overall process for the proposed fast reverse engineering method. For a 4-input LUT, each LUT is instantiated with an individual internal variable $A_j$ ($1 \leq j \leq N$) by modifying XDL [34,35], and 16-bit **BIT** for each individual internal variable $A_j$ ($1 \leq j \leq N$) is generated. Since the pairs of ($I_0$, $A_3$), ($I_1$, $A_4$), ($I_2$, $A_1$), ($I_3$, $A_2$) are discovered, 4 **BIT** vectors associated with $A_j$ ($1 \leq j \leq N$) are allocated at the mapping table in the order of $A_2$, $A_1$, $A_4$, and $A_3$. Consequently, Figure 6 shows the mapping table computed from the proposed fast method. Note that it is identical to the mapping table computed from the previous exhaustive method in Figure 4. Since the same mapping table is obtained in both previous and proposed methods, the same results are guaranteed. To sum up, we tried to obtain 16-bit **BIT** associated with each input variable $I_i$ ($0 \leq i < N$) to reduce processing complexity from exponential increase to linear increase at the first time. However, the practical issue prohibits from instantiating an LUT with a single variable $I_i$ ($0 \leq i < N$), so we implement a LUT with a single internal variable $A_j$ ($1 \leq j \leq N$) using the help of XDL. First, an *I–A* mapping table is constructed, and **BIT** vectors corresponding to the internal variable $A_j$ ($1 \leq j \leq N$) are generated. Lastly, the **INIT-BIT** mapping table is constructed using *I-A* relation and **BIT** vectors. The proposed method requires $N$ processing to build a mapping table between $I_i$ ($0 \leq i < N$) and $A_j$ ($1 \leq j \leq N$) and additional $N$ processing for building **BIT** vectors associated with $A_j$ ($1 \leq j \leq N$).

## 4. Experimental Results

　　Since the final mapping tables between **INIT-BIT** are the same as shown in Figures 4 and 6, the previous and proposed methods always provide the same reverse outputs. To compare two methods in terms of recovery time, we measure the total processing time to restore a target function from a bitstream. The previous and proposed methods can be applied to all Xilinx FPGAs, which are synthesized and implemented with Xilinx ISE Design Suite. As an example, low-end FPGA of Xilinx Spartan-3 equipped with 4-input LUTs and high-end FPGA of Xilinx Virtex-5 equipped with 6-input LUTs are synthesized, and Xilinx ISE Design Suite v10.1 is used to generated BIT vectors under 3.7 GHz Intel Core i5 with 16 G RAM in the experiments. Table 1 shows the details of recovery time for a single LUT restoration. Both methods consist of three steps: bit position search, mapping table construction, and logic restoration, and the proposed method uses the same bit position search and logic restoration but improves mapping table construction compared with the previous method. As described in the previous sections, the proposed method saves the processing from $2^N$ to $2N$ and results in significant improvement in mapping table construction. In addition, mapping table construction contains the majority of recovery time, and therefore the overall reduction from the proposed method is significant for both Xilinx Sparatn-3 and Virtex-5. According to Table 1, the proposed method saves recovery time by 57% and 86% for Xilinx Spartan-3 and Virtex-5, respectively.

　　For a practical comparison, various benchmarks and real cryptography applications are implemented in Xilinx Spartan-3 and Virtex-5. Table 2 shows the number of utilized LUTs and total recovery time according to reverse engineering methods. Five designs from ISCAS'85 benchmarks are implemented, whose LUTs ranges from 2 to 703. As practical application, data encryption standard (DES) and advanced encryption standard (AES) are implemented, whose LUTs ranges from 983 to 7644. It is noticeable that the number of the utilized LUTs differs depending on Xilinx FPGA due to the fact that a different FPGA includes difference internal hardware resources. For example, when the DES circuit is synthesized and implemented, Xilinx Spartan-3 requires 1397 4-input LUTs and Xilinx

Virtex-5 requires 983 6-input LUT LUTs. Significant improvement is expected in Table 2, since the total recovery time is proportional to the unit recovery time for a single LUT. According to Table 2, the proposed method shows superior recovery time through all comparisons, given the bitstream generated from the identical design. For instance, when the DES circuit is recovered, the proposed method saves 57% in Xilinx Spartan-3 equipped with 4-input LUT and 85% in Xilinx Virtex-5 equipped with 6-input LUT compared to the previous exhaustive method. It is noticeable that the reduction time becomes more significant as the number of input increases.

**Table 1.** Comparison of recovery time for a single LUT.

| FPGA Type (LUT Type) | Spartan-3 (4-Input LUT) | | Virtex-5 (6-Input LUT) | |
|---|---|---|---|---|
| Algorithm | Exhaustive [6] | Proposed | Exhaustive [6] | Proposed |
| Bit position search | 1.1 min | 1.1 min (0%) | 1.6 min | 1.6 min (0%) |
| Mapping table construction | 8.5 min | 3.1 min (64%) | 52 min | 6.3 min (88%) |
| Logic restoration | 0.01 ms | 0.01 ms (0%) | 5 ms | 5 ms (0%) |
| Total time | 9.6 min | 4.1 min (57%) | 53.6 min | 7.9 min (86%) |

**Table 2.** Comparison of recovery time for ISCAS'85 benchmarks and practical applications.

| FPGA (LUT Type) | | Spartan-3 (4-Input LUT) | | | Virtex-5 (6-Input LUT) | | |
|---|---|---|---|---|---|---|---|
| Algorithm | | # of LUT | Time | | # of LUT | Time | |
| | | | Exhaustive [6] | Proposed | | Exhaustive [6] | Proposed |
| ISCAS'85 Benchmark | C17 | 2 | 19.2 min | 8.3 min | 2 | 1.8 h | 15.9 min |
| | C499 | 78 | 13.1 h | 5.7 h | 66 | 2.5 days | 8.7 h |
| | C880 | 115 | 18.6 h | 8.0 h | 77 | 2.9 days | 10.2 h |
| | C1908 | 103 | 16.2 h | 6.9 h | 81 | 3.0 days | 10.7 h |
| | C3540 | 326 | 2.2 days | 0.9 days | 222 | 8.3 days | 1.2 days |
| | C6288 | 703 | 4.7 days | 2.0 days | 468 | 2.5 weeks | 2.6 days |
| Practical applications | DES | 1397 | 9.3 days | 4.0 days | 983 | 5.2 weeks | 5.4 days |
| | AES | 7644 | 7.3 weeks | 3.1 weeks | 2742 | 14.7 weeks | 2.2 weeks |

## 5. Conclusions

This paper presents a novel logic extraction method from Xilinx FPGA bitstreams. Whereas the previous method demands exhaustive $2^N$ processing for a *N*-input LUT, the proposed method reduces the processing to $2N$. In experimental results, Xilinx Spartan-3 equipped with 4-input LUTs and Virtex-5 equipped with 6-input LUTs are utilized for a fair comparison. Various designs associated with ISCAS'85 benchmarks and cryptography applications are implemented, and all the results show that the proposed method outperforms the previous method. According to the experimental results, the proposed method can save 57% and 86% recovery time compared to the previous method. The improvement becomes more significant in future Xilinx FPGAs as the commercial Xilinx FPGAs tend to include LUTs with more inputs. Our next research aim is to study a fast reversing method for high-end FPGA chips using Xilinx Vivado that uses more complex obfuscation compared to Xilinx ISE Design Suite.

**Author Contributions:** Conceptualization, H.Y.; methodology, S.C.; software, S.C.; validation, S.C. and H.Y.; formal analysis, H.Y.; investigation, S.C.; resources, S.C.; data curation, S.C.; writing—original draft preparation, S.C.; writing—review and editing, H.Y.; visualization, S.C.; supervision, H.Y.; project administration, H.Y.; funding acquisition, H.Y. All authors have read and agreed to the published version of the manuscript.

**Funding:** This research funded by research fund of Chungnam National University.

**Acknowledgments:** This work was supported by research fund of Chungnam National University.

**Conflicts of Interest:** The authors declare no conflict of interest.

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
