# Peer review of "Fast Logic Function Extraction of LUT from Bitstream in Xilinx FPGA"

_electronics, doi:10.3390/electronics9071132_

Round 1

Reviewer 1 Report

The paper shows improved reverse engineering method for extracting the LUT configuration in the FPGAs. In general, the paper is well written, the plots are high-quality and relatively clear to read. Please find my remarks below:

line 36 - for completeness please include FPGAs with encrypted bitstream.
line 42 - "recently" - reverse engineering of FPGAs is as old as FPGAs themselves, please rephrase or emphasize the "trojan" topic, which is indeed a new one, or at least recently heavily published.
line 116 - "privious" ?

This paper also discusses relatively old architectures and old synthesis tool which makes everything outdated. Feasibility tests with newer architectures would be very beneficial. In particular, that Vivado and modern FPGAs could include other mechanisms of obfuscating the true content of the configuration. 

Moreover, could you describe an ethical use of reverse engineering methodologies, in particular the one described in this paper? 

Reviewer 2 Report

This work presents an extraction method specific for Xilinx FPGA bitstreams. The authors managed to demonstrate a reduction of extraction time relative to previous extraction methods.

The proposed method is specific for Xilinx Sparrtan and Virtex 5 implementations. This research has very limited scientific scale and there is no discussion for general applicability. Basically it appears like an engineering solution to a specific implementation. Still the author’s background description is eloquently written and I suppose that it has some merit to reverse engineering researchers.

Reviewer 3 Report

I enjoyed reading this paper. The presentation of the method is quite clear and the experimental results show large speedups over the state-of-the-art. I have some suggestions to improve the manuscript.

First, the work is not well motivated in the introduction. In my opinion, it is not obvious why reverse engineering the bitstream is relevant for the research community. Authors should discuss the security issues and concerns, as previous papers do.

Second, authors overlooked some relevant recent work in the area, such as:

Yoo, Ho Young, So Yeon Choi, and Ji Woon Park. "Reverse Engineering for Xilinx FPGA Chips using ISE Design Tools." Journal of Integrated Circuits and Systems 6.1 (2020).

Finally, please review the text to fix spelling mistakes and typos, and remove the repeated sentence in page 6.

Round 2

Reviewer 1 Report

Dear Authors, thank you for taking into account my comments.